# RSL3 Inhibits Porcine Epidemic Diarrhea Virus Replication by Activating Ferroptosis

**DOI:** 10.3390/v15102080

**Published:** 2023-10-12

**Authors:** Yingguang Li, Yuwei Bao, Yan Li, Xiaoxiao Duan, Shaoming Dong, Jiaxu Lin, Xiaoyun Chang, Yue Tan, Hongliang Zhang, Hu Shan

**Affiliations:** 1College of Veterinary Medicine, Shandong Collaborative Innovation Center for Development of Veterinary Pharmaceuticals, Qingdao Agricultural University, Qingdao 266109, China; 20202113026@stu.qau.edu.cn (Y.L.); baoyuwei@stu.qau.edu.cn (Y.B.); dawnyeah2023@163.com (S.D.); mzfdc8@163.com (J.L.); cxy17806259780@163.com (X.C.); 17863662503@163.com (Y.T.); 199001007@qau.edu.cn (H.S.); 2Qingdao Animal Disease Prevention and Control Center, Qingdao 266100, China; liyanqd2008@163.com (Y.L.); dxx45@163.com (X.D.)

**Keywords:** (1S,3R)-RSL3, GPX4, porcine epidemic diarrhea virus, ferroptosis

## Abstract

Porcine epidemic diarrhea virus (PEDV) is a highly contagious coronavirus that induces diarrhea and death in neonatal piglets, resulting in substantial economic losses to the global swine industry. The mechanisms of PEDV infection and the roles of host factors are still under exploration. In this study, we used the ferroptosis pathway downstream target activator (1S,3R)-RSL3 compound as a starting point, combined with the interactions of N-acetylcysteine and deferoxamine, to elucidate the effects of a series of compounds on PEDV proliferation. We also established glutathione peroxidase 4 (GPX4) gene overexpression to further elucidate the relationship between the ferroptosis pathway and PEDV. (1S,3R)-RSL3 inhibited PEDV replication in Vero cells, while N-acetylcysteine and deferoxamine promoted its proliferation. In addition, (1S,3R)-RSL3 mainly affected the replication stage of PEDV. Overexpression of GPX4 promoted PEDV proliferation, indicating that the ferroptosis pathway could influence PEDV replication in Vero cells. This study focused on the mechanism of (1S,3R)-RSL3 inhibition on PEDV, laying the foundation for exploring the pathogenic mechanisms of PEDV and drug development.

## 1. Introduction

Porcine epidemic diarrhea virus (PEDV) is a highly contagious coronavirus that causes severe diarrhea and mortality in neonatal piglets, resulting in substantial economic impacts on the swine industry worldwide [1,2]. PEDV is transmitted via the fecal–oral route and causes atrophy and shedding of small intestinal villi, resulting in severe diarrhea, vomiting and dehydration [3]. PEDV is distributed in many countries and regions including the United States, Europe and Asia, but PEDV in China exhibits higher recombination and evolution rates with increased risks of cross-species transmission [4,5]. Recent epidemiological studies in southern Italy reveal variability in PEDV epidemiology and transmission risks between different animal populations [6,7]. In domestically raised pigs in intensive farming systems, PEDV seroprevalence was much higher than in wild boars in the same region, reaching 14.8%, and showed an increasing trend in adult pigs [7,8]. This poses challenges for research on how to control the spread of this infectious disease. To reveal PEDV pathogenesis, interactions between PEDV and important host signaling pathways, like reactive oxygen species (ROS), have been further studied by researchers [9,10,11].

Ferroptosis is a form of iron-dependent regulated cell death. It is characterized by disruption of the lipid repair system mediated by glutathione peroxidase 4 (GPX4), which converts toxic lipid peroxides to harmless lipid alcohols. When lipid peroxide levels exceed GPX4’s enzymatic capacity, lipid ROS accumulate and cause oxidant damage to cellular components [12,13,14]. (1S,3R)-RSL3 targets and inhibits GPX4, leading to ferroptotic cell death in sensitive cells unable to eliminate accumulated lipid peroxides [15]. GPX4 is one of the key influencing factors in the replication process of some viruses. GPX4 deficiency can suppress innate antiviral immune responses induced by herpes simplex virus-1 (HSV-1), promoting HSV-1 replication in vivo. Clinical studies on iron metabolism in SARS-CoV-2 infection show abnormal iron metabolism in COVID-19 patients, with initially decreased then increased serum iron levels after treatment. Newborn piglets, especially iron-deficient ones, are more susceptible to PEDV, and supplementation with ferrous citrate damages PEDV infection in vivo and in vitro [16,17,18]. Recent studies have shown that lauric acid (LA) and salinomycin stimulate ROS production and inhibit PEDV replication, and both compounds are known to stimulate ROS production in cells, which is ones of main characteristic of ferroptosis [19,20]. Research confirms that medium chain fatty acids can affect the infectivity of PEDV [21]. In previous studies, Erastin as a classical ferroptosis activator was also found to inhibit viral replication [22]. It is still unclear which regulatory pathways Erastin utilizes to achieve viral replication inhibition, but verification of this effect from RSL3 could further elucidate whether modulation of the System xc cystine transporter (cystine synthesizes glutathione (GSH)) pathway affects PEDV.

In this study, we take RSL3 compound as a starting point, combining the interaction between DFOM and NAC, to elucidate the effects of a series of compounds on PEDV proliferation. As a downstream target activator of the Ferroptosis pathway, this study will further explore the impact of this pathway on PEDV proliferation, and extend the presentation of this conclusion through antioxidant experiments, IFA and other methods. This study contributes to the understanding of the mechanism of PEDV infection and provides a theoretical basis for the development of antiviral drugs.

## 2. Materials and Methods

### 2.1. Cells, Viruses and Reagents

Vero cells were obtained from Shandong Province Key Laboratory of Preventive Veterinary Medicine and cultured in high glucose Dulbecco’s Modified Eagle Medium (DMEM; HyClone Laboratories, Inc., Logan, UT, USA) supplemented with 10% fetal bovine serum (VivaCell, Shanghai, China). We used the cells within 20 passages. Experiments were conducted when monolayer confluence reached 90%. All cell culture was performed at 37 °C in a 5% CO_2_ incubator. The PEDV SD2020 strain (GenBank ID: OP894120) was provided by Shandong Province Key Laboratory of Preventive Veterinary Medicine. The virus infection maintenance medium consisted of serum-free high glucose DMEM with added 12.5 μg/mL trypsin (Gibco, Carlsbad, CA, USA). Briefly, confluent monolayers were washed once with D-PBS, infected with PEDV in maintenance medium containing 12.5 μg/mL trypsin for 1 h at 37 °C. Virus inoculum was removed and cells were washed twice with PBS before adding back to maintenance medium. Rabbit polyclonal anti-PEDV-S antibody was prepared in-house by our laboratory. FITC-conjugated goat anti-rabbit IgG antibody and HRP-conjugated rabbit anti-His antibody were obtained from Shenyang Wanlei Bioscience Co., Ltd. (Shenyang, China). (1S,3R)-RSL3, acetylcysteine (NAC), and deferoxamine mesylate (DFOM) were provided by Topscience Co. Ltd. (Shanghai, China). The GPX4-6*His fusion expression plasmid was synthesized by Sangon Biotech Co., Ltd. (Shanghai, China).

### 2.2. Cell Viability Assay

The Cell Counting Kit-8 (CCK-8) (Beyotime Biotech, Shanghai, China) was utilized to quantify the corresponding optical density ratio marker cell viability levels in 96-well plates as per the manufacturer’s guidelines, with CCK-8 assessing the cytotoxicity of RSL3, DFOM and NAC through cell vitality evaluation. To assess cell viability, Vero cells were resuspended at 1 × 10^4^ cells/mL and 100 mL was transferred into 96-well plates, preincubated (37 °C, 5% CO_2_) for 12 h before adding RSL3 (0.5–8 μM), DFOM (10–50 μM) and NAC (5–85 μM); after 18 h incubation, 10 mL of CCK-8 solution was added to each well, incubated for 2 h protected from light, then cell viability levels were calculated by measuring absorbance at 450 nm using a microplate reader (Tecan, Mannedorf, Switzerland) [19].

### 2.3. Lipid Peroxidation Assay

Lipid peroxidation assay was used to detect the biological activity of compounds in Vero cells. Lipid peroxidation levels were determined by measuring MDA levels according to the manufacturer’s guidelines (Beyotime Biotech, China). After 12 h of treatment, cells were washed three times with PBS and homogenized supernatants by lysing cells with NP-40 (Beyotime Biotech, China). Then, the TBA-MDA mixture was heated to 100 °C for 60 min. Absorbance of the mixture was measured at 532 nm by enzymatic labeling, and MDA concentrations were calculated as Μm/mg protein based on the standard curve [23].

### 2.4. Quantitative Real-Time PCR

Total RNA was extracted using the SteadyPure Universal RNA Extraction Kit (Accurate Biotechnology Co., Ltd., Changsha, China) as per manufacturer protocol. Reverse transcription to synthesize first-strand Cdna was performed with HiScript^®^ II Q RT SuperMix (Vazyme Biotech Co., Ltd., China). SYBR Green Pro Taq HS Premix Qpcr Kit (Accurate Biotechnology Co., Ltd.) was used for fluorescent PCR with the Bio-Rad CFX Connect System (Applied BiosystemsTM Quant Studio 5). The 20 mL qRT-PCR reaction comprised 10 mL 2× SYBR Green Premix, 1 mL each forward and reverse primer (10 pmol/L), 2 mL template DNA and 6 mL sterile water (primer sequences in Table 1). Thermocycling conditions were: 30 s denaturation, then 40 cycles at 95 °C for 30 s, 60 °C for 30 s, 72 °C for 30 s, followed by melting curve analysis. Relative quantification (2^−ΔΔCt^) calculations were performed [24,25].

### 2.5. Indirect Immunofluorescence

For indirect immunofluorescence, cells were cultured in 48-well plates, grouped and treated. After rinsing with PBS, cells were fixed with 4% formaldehyde for 30 min and washed 3 times with PBS. Cells were permeabilized on ice using 0.1% Triton X-100 for 30 min and washed 3 times with PBS. Cells were blocked using Immunofluorescence Fast Blocking Solution for 30 min, then incubated for 3 h with rabbit anti-PEDV prickle protein polyclonal antibody. This was followed by incubation with FITC goat anti-rabbit antibody for 1 h at 37 °C. Cells were washed 3 times with PBS, stained with DAPI for 5 min, blocked with glycerol, and visualized by fluorescence microscopy [25].

### 2.6. Detection of Reactive Oxygen Species

ROS (reactive oxygen species) levels were quantified using the Beyotime ROS assay kit (Beyotime Biotechnology, China). Briefly, Vero cells at 1 × 10^4^ density in black 96-well plates were treated with 1 Mol PEDV, RSL3 (3 μM), deferoxamine (30 μM), acetylcysteine (15 μM), and control groups for 6 h. Samples were collected per kit protocol, cells were washed twice with PBS and incubated with 10 μM DCFH-DA (2′,7′-dichlorofluorescin diacetate) for 30 min at 37 °C. After washing thrice with serum-free medium, fluorescence intensity was detected at 488 nm excitation and 525 nm emission wavelengths using a microplate reader to determine changes in intracellular ROS levels compared to control [19].

### 2.7. Western Blot

In order to validate the expression of GPX4-His fusion protein, Vero cell lysates from 6-well plates were lysed with NP-40 buffer. A small lysate sample was taken for protein quantification to determine concentration. Protein loading amounts were determined based on concentration, and 1× loading buffer was added. For sample denaturation, lysates in buffer were boiled for 7 min then cooled on ice. Protein samples and marker were loaded into Sodium dodecyl sulfate polyacrylamide gel electrophoresis (SDS-PAGE) gel wells and electrophoresed at 120 V for 30 min, then at 90 V for 1 h. Polyvinylidene difluoride (PVDF) membranes were activated in methanol for 5 s. Gels were pried open, trimmed to size, and transferred onto wet filter paper. The transfer cassette was assembled and proteins transferred at 200 mA for 60 min. PVDF membranes were blocked in 5% milk in Tris-buffered saline with Tween 20 (TBST) at 37 °C for 2 h with gentle shaking. Membranes were washed in TBST, then probed with rabbit anti-His monoclonal antibody (1:5000) overnight at 4 °C with gentle shaking. After washing with TBST, membranes were incubated with HRP goat anti-rabbit secondary antibody for 1 h at room temperature. After removing secondary antibody, membranes were washed 3 times with TBST for 5 min each. Finally, Protein expression levels were assessed by immunoblotting using ECL reagent with band intensities analyzed by ImageJ1.52a software.

### 2.8. Statistical Analysis

All data are expressed as the mean (SD) of at least three independent experiments. Statistical analyses were performed using GraphPad Prism 6.0 software and Microsoft Excel 2016. Differences among multiple groups were analyzed by ordinary one-way ANOVA followed by comparisons between each two groups. *p* values less than 0.05 were considered statistically significant. Statistical significance is indicated by * for *p* < 0.05, ** for *p* < 0.01 in the figures and tables.

## 3. Results

### 3.1. Evaluation of the Cytotoxicity of (1S,3R)-RSL3, Deferoxine Mestlate and NAC

To investigate the potential cytotoxicity of RSL3 and deferoxamine mesylate, CCK-8 assays were performed to assess cell viability of Vero cells treated with varying concentrations of these compounds for 16 h. RSL3 significantly decreased cell viability at 6 μM compared to the control group (Figure 1A). Deferoxamine mesylate significantly reduced cell viability at concentrations of 35 μM and above (Figure 1B). NAC at concentrations from 5 to 85 μM did not exhibit significant cytotoxicity compared to the control group (Figure 1C). Therefore, subsequent experiments using RSL3 were performed at concentrations below 5 μM, Deferoxamine mesylate at 30 μM, and NAC at 15 μM as a routine usage concentration.

### 3.2. Validation of the Effects of RSL3, NAC and DFOM on Lipid Oxidation in Vero Cells

RSL3 can inhibit GPX4 leading to lipid peroxidation accumulation and inducing ferroptosis, while NAC and DFOM can inhibit the progression of ferroptosis. This experiment validated the biological activity of the compounds (Figure 2A). DFOM (30 μM) significantly inhibited intracellular MDA content (*p* < 0.01). Compared to the RSL3 (3 μM) + NAC (15 μM) co-treatment group and the RSL3 (3 μM) + DFOM (30 μM) co-treatment group, the MDA content significantly increased in the sole RSL3 treatment group (*p* < 0.01). In the statistical results, we observed that the MDA content of the RSL3 group showed no significant difference compared to the Vero group, but the comparisons with the co-treatment groups demonstrated that the compounds exerted their respective biological functions in Vero cells. 

### 3.3. Effects of RSL3, NAC and DFOM on PEDV Proliferation in Vero Cells

In this study, we investigated the effect of RSL3, NAC and DFOM on PEDV proliferation in Vero cells. The results of fluorescent PCR showed that, compared with PEDV-infected Vero cells, RSL3 significantly inhibited PEDV proliferation at 6 h (*p* < 0.05), while NAC and DFOM promoted PEDV proliferation (*p* < 0.01) (Figure 2B). The co-treatment of compounds showed that the inhibitory effect of RSL3 on PEDV proliferation was neutralized by NAC and DFOM (*p* < 0.0001). The fluorescent PCR results confirmed the antiviral effect of RSL3 and the promoting effect of NAC and DFOM.

Indirect immunofluorescence results (Figure 3A) showed that NAC promoted PEDV propagation and cytopathic effect (CPE) formation, while RSL3 inhibited viral proliferation. Fluorescence intensity quantification using ImageJ1.52a software was utilized to quantify the visual observations (Figure 3B). Taken together, the data demonstrate that RSL3, NAC and DFOM significantly modulated PEDV proliferation. Specifically, RSL3 markedly suppressed PEDV propagation and CPE formation. Furthermore, RSL3 exhibited an effective inhibitory effect on NAC- and DFOM-induced PEDV proliferation.

### 3.4. RSL3 Affects the Adhesion, Invasion, Replication and Release of PEDV on Vero Cells

To explore the antiviral mechanism of RSL3 against PEDV infection in Vero cells, we assessed its effect on the adhesion and invasion stages of the viral replication cycle (Figure 4A). We measured the intracellular viral RNA levels by RT-qPCR after treating the cells with RSL3 at each stage. Interestingly, our results showed that RSL3 did not impact the adhesion and invasion of PEDV into Vero cells (Figure 4B). To determine the effect of RSL3 on PEDV replication and release, we performed two assays (Figure 4A). For the replication assay, we first infected Vero cells with PEDV and then treated them with RSL3 + PEDV or PEDV. At 5 h post-infection, we extracted the total intracellular RNA and measured the viral RNA levels by RT-qPCR. The results showed that RSL3 + PEDV significantly reduced PEDV replication in Vero cells (*p* < 0.05). For the release assay, we first infected Vero cells with PEDV and cultured them for 5 h, then replaced the medium with RSL3-supplemented or control medium for 3 h. We collected the supernatant and the pellet and extracted the viral RNA from both fractions. We assessed the release of PEDV by comparing the viral RNA levels in the supernatant and the content of un-released PEDV in the cells by comparing the viral RNA levels in the pellet (Figure 4A). The results showed that RSL3 had no effect on PEDV release from Vero cells, as there was no significant difference in the viral RNA levels between the treatment and control groups in the supernatant, while the intracellular PEDV RNA showed a significant down-regulation, which may be related to RSL3′s ability to inhibit viral replication. This indicates that RSL3 mainly affects PEDV replication rather than adhesion, invasion and release, and through the release experiment we found that this inhibitory effect is rapid and limited. It does not prevent Vero cell death.

### 3.5. Effects of RSL3, NAC, DFOM and PEDV Interactions on ROS

Reactive oxygen species (ROS) accumulation is a prerequisite for ferroptosis induction. To evaluate interactions between the compounds and PEDV, ROS levels were assessed (Figure 5A). The results showed that RSL3 induced higher ROS accumulation compared to NAC (*p* < 0.05). PEDV alone also stimulated ROS production (*p* < 0.05). Moreover, co-treatment with RSL3 and PEDV led to further significant ROS accumulation compared to RSL3 alone (*p* < 0.05). Due to the antioxidant effect of NAC, PEDV with RSL3 co-treatment accumulated markedly higher ROS levels than PEDV with NAC co-treatment (*p* < 0.001). Despite this, the RSL3 and PEDV with NAC co-treatment group still upregulated ROS versus control. These data re-confirmed the activating effect of PEDV on ROS. Both PEDV-induced ROS and overall ROS generation could be suppressed by NAC and DFOM. Interestingly, DFOM also inhibited PEDV-triggered ROS. Although the ROS results seem contradictory to RSL3-induced ferroptosis inhibiting PEDV, we speculate based on our validations that PEDV sensitivity to ROS is a dynamic equilibrium that can be modulated through multiple pathways.

### 3.6. Overexpression of GPX4-His Affects the Interaction between PEDV and RSL3 during Replication

To verify the effect of GPX4 overexpression on RSL3 inhibition of PEDV, we constructed a recombinant pcDNA3.1-GPX4-His plasmid (Figure 5B) and transfected it into Vero cells to observe the replication kinetics of PEDV in the presence or absence of GPX4, and verified the regulation of GPX4-His protein by RSL3 during the PEDV replication process by Western Blot. After *Hind* III and *EcoR* I double digestion of the recombinant plasmid, we obtained a 531 bp GPX4-His band, which proved that the plasmid was correct (Figure 5C).

From the schematic diagram in Figure 6A, we can understand the important position of GPX4 gene in the ferroptosis pathway, and RSL3 is also an inhibitor targeting GPX4. By proving the regulation of GPX4, we can determine whether RSL3 inhibits PEDV by inhibiting GPX4. The results of fluorescence PCR showed that PEDV significantly upregulated the expression of GPX4 gene, while RSL3 inhibited the expression of GPX4 gene (*p* < 0.05) (Figure 6B). The same nucleic acid samples were used to detect the PEDV replication level, and the results showed that GPX4 significantly promoted PEDV replication, while RSL3 significantly inhibited PEDV replication in both GPX4-overexpressing and non-overexpressing groups (*p* < 0.001), and there was no significant difference between the two groups (Figure 6C). Western Blot detected the expression level of GPX4-His fusion protein (Figure 6D), and it was observed that PEDV upregulated GPX4 gene, while RSL3 downregulated the expression of fusion protein. This result was consistent with Figure 6B, which proved that RSL3 downregulated GPX4 gene in the process of inhibiting PEDV replication, and at the same time GPX4 overexpression enhanced the antioxidant effect and promoted virus proliferation in Vero cells.

## 4. Discussion

Due to frequent mutations and limitations of susceptible host cell lines, research on the mechanisms of PEDV infection and functional host factors is still improving [16]. The host factors required to maintain coronavirus replication are major targets for antiviral drugs. During infection, viruses often utilize components of the host cell to facilitate viral entry, replication, and assembly. Resisting viral invasion by conferring protection from infection on the host cell, these factors are often referred to as antiviral host factors. Antiviral factors are positive regulators of the innate immune system and become potential targets for antiviral therapy [26]. ROS plays complex roles in viral infection and antiviral therapy, depending not only on the type, amount and location of ROS, but also on the type of virus, its life cycle and adaptability. Some viruses can utilize ROS to promote their own replication and expression, such as human papillomavirus (HPV) and dengue virus (DENV) [27,28,29] These viruses can induce host cells to produce excessive ROS, thus accelerating cell cycle progression, inhibiting host antiviral responses, and inducing DNA damage and mutation. To maintain ROS levels, these viruses encode their own or induce host cells to express antioxidant enzymes and antioxidants. On the other hand, some viruses are sensitive to ROS and inhibited by them, such as influenza A virus (IAV) and hepatitis B virus (HBV) [30,31,32]. Infection by these viruses can induce host cells to produce excessive ROS, thus damaging viral RNA/DNA and proteins, inhibiting their replication and assembly. At the same time, ROS can also activate host antiviral responses such as interferon and other antiviral factors [32,33,34]. In PEDV, ROS and P53 can inhibit apoptosis in cells, and can activate the PERK/ROS axis to maintain replication in Vero cells [10,11]. Given the important roles of ROS in viral infection, we found that some drugs inhibited viral replication by activating ROS production or enhancing ROS effects, displaying certain antiviral activities. For example, sodium valproate can increase intracellular ROS levels by activating complex I and III in the mitochondrial respiratory chain, thus inhibiting HPV replication and expression [35]. Sodium thiosulfate can produce excessive ROS by activating NADPH oxidase, thus inhibiting IAV infection and replication [33]. In this study, we found the mutual regulation of PEDV and ROS to be perplexing as it did not align with the expected results of regulation induced by ferroptosis compounds. The use of iron death activates accumulation of lipid peroxidation in cells, but this result shows an inhibitory effect on PEDV proliferation in the experiment (Figure 6). With reference to the aforementioned content on the interaction between ROS and viruses, we hypothesize that PEDV proliferation in Vero cells is sensitive to and regulated through multiple pathways of ROS homeostasis. PEDV infection itself activates innate immunity leading to ROS accumulation, while active induction of ferroptosis can inhibit PEDV propagation.

In previous studies, DFOM has been shown to inhibit the Ferroptosis induced by ferric ammonium citrate (FAC), and promote viral load in the intestines of piglets infected with PEDV [18]. In this study, we reproduced similar in vitro results of DFOM promoting PEDV proliferation in Vero cells, and confirmed that it can interfere with the inhibitory effect of RSL3 on PEDV replication [18,36]. In our observations, NAC produced enhancement of PEDV proliferation, similar to DFOM, which was evident from IFA observations (Figure 3A). This result is controversial in some reports. NAC as a classic antioxidant affecting PEDV proliferation has been reported. In some articles, NAC did not significantly inhibit PEDV. In the report of Qian Zhang et al., NAC alleviated PEDV diarrhea by regulating intestinal reactions but promoted replication by interfering with interferon signaling pathways [20,37,38]. In this experiment, we verified that the influence stage of RSL3 is also the replication stage (Figure 4C), and NAC can inhibit the inhibitory effect of RSL3, which is consistent with the above research results. 

In this study, we demonstrated that RSL3 inhibits PEDV replication by regulating GPX4 to induce ferroptosis and modulate intracellular lipid and ROS levels. One confounding point in our study was the observation that RSL3 inhibition of PEDV occurred more rapidly and prominently than its induction of intracellular ROS accumulation. We hypothesized two explanations for this phenomenon. First, the localized regulation of PEDV through the ferroptosis pathway may involve more than just ROS, and potentially other more significant pathways. In existing research, both lipids and iron have been shown to significantly impact PEDV replication, so further in-depth research is still needed on exactly how ferroptosis progression affects PEDV proliferation [18,21,39,40]. Second, RSL3 inhibition of PEDV may rely primarily on unknown pharmacodynamic effects rather than targeting GPX4. From our literature review, RSL3 research has focused on ferroptosis caused by GPX3 and GPX4, while ferroptosis occurrence is often synergistic with autophagy [40,41]. The distinction between autophagy and ferroptosis mechanisms in PEDV research needs clarification. We demonstrated RSL3 inhibition of PEDV replication in Vero cells in vitro, laying the foundation for exploration of the innate immune mechanisms of PEDV and developing antiviral drugs.

## 5. Conclusions

RSL3 inhibits PEDV replication in Vero cells and does not significantly affect the adhesion, invasion, and release stages. In contrast, NAC and DFOM can promote PEDV proliferation in Vero cells. Further verification of GPX4 gene regulation showed that RSL3 can inhibit GPX4 and upregulate ROS, while GPX4 overexpression can promote PEDV proliferation. This demonstrates that the ferroptosis pathway can influence PEDV replication in Vero cells. This study focused on investigating the mechanism of RSL3 in inhibiting PEDV, which lays the foundation for exploring the pathogenic mechanisms of PEDV.

## Figures and Tables

**Figure 1 viruses-15-02080-f001:**
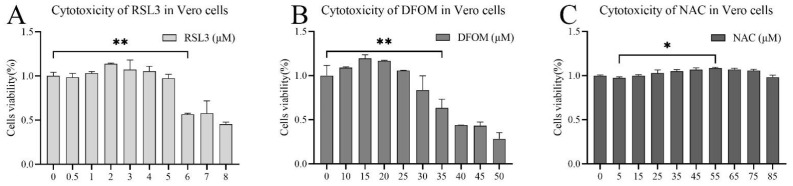
Effects of RSL3, Deferoxamine mesylate, and NAC on Vero cell viability. Vero cells were treated with different concentrations of RSL3, Deferoxamine mesylate, and NAC for 16 h, and cell viability was measured by CCK-8 assay. (**A**) Effect of RSL3 on Vero cell viability. (**B**) Effect of Deferoxamine mesylate on Vero cell viability. (**C**) Effect of NAC on Vero cell viability. The data were taken from three independent experiments, with the untreated group as the control group. The differences were evaluated using ANOVA test. The error bar represents the standard deviation, * *p* < 0.05, ** *p* < 0.01.

**Figure 2 viruses-15-02080-f002:**
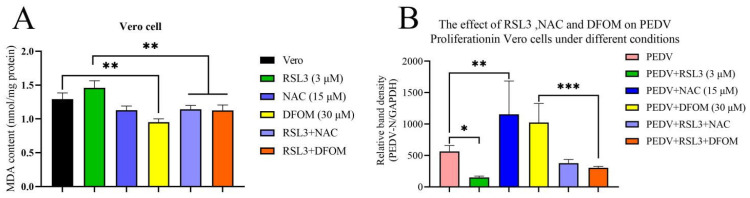
Effects of RSL3, NAC and DFOM on lipid peroxidation and PEDV proliferation in Vero cells. (**A**) MDA content in Vero cells treated with RSL3, NAC and DFOM for 12 h. (**B**) PEDV titer in Vero cells infected with PEDV and treated with RSL3, NAC and DFOM for 6 h. The data were taken from three independent experiments, with the untreated group as the control group. The differences were evaluated using ANOVA test. The error bar represents the standard deviation, * *p* < 0.05, ** *p* < 0.01, *** *p* < 0.001 compared with the corresponding control group.

**Figure 3 viruses-15-02080-f003:**
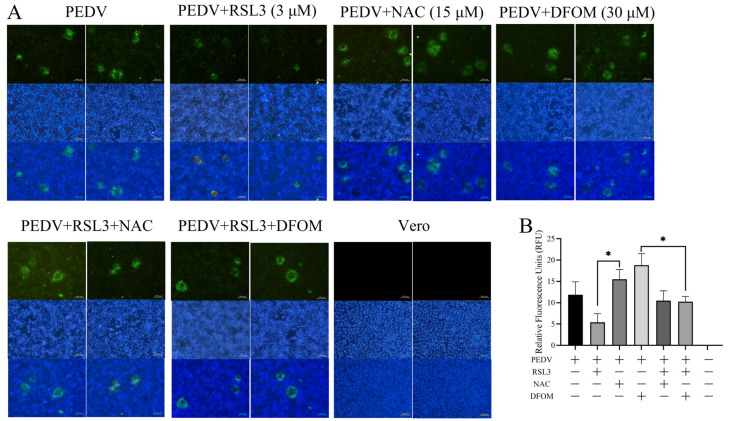
Indirect immunofluorescence observation of the effects of RSL3, NAC and DFOM on PEDV-induced CPE in Vero cells. (**A**) Immunofluorescence staining of PEDV N protein (green) and DAPI (blue) in Vero cells infected with PEDV and treated with RSL3, NAC and DFOM for 6 h. (**B**) Quantification of fluorescence intensity of PEDV N protein in Vero cells using ImageJ1.52a software. The differences were evaluated using ANOVA test. The error bar represents the standard deviation, * *p* < 0.05.

**Figure 4 viruses-15-02080-f004:**
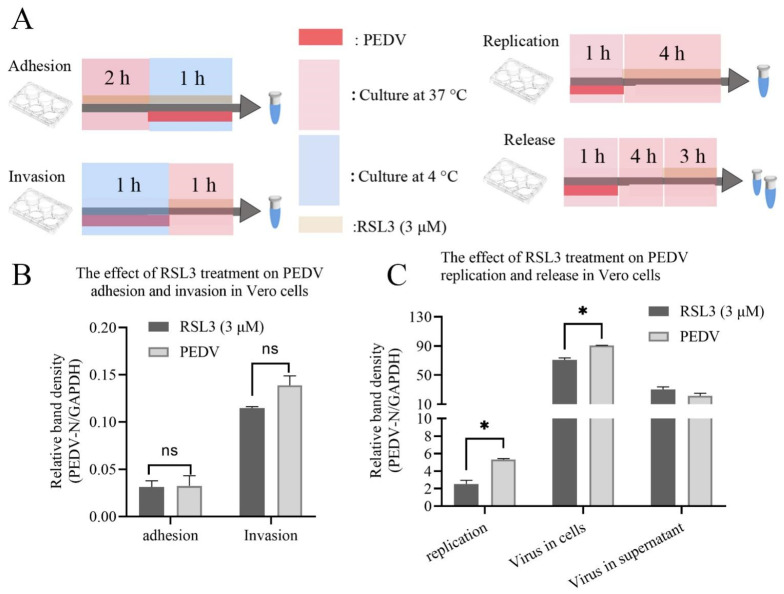
Effect of RSL3 on different stages of PEDV infection in Vero cells. (**A**) Schematic diagram of the experimental design. Vero cells were treated with RSL3 at different stages of PEDV infection: adhesion, invasion, replication, and release. (**B**) Relative viral RNA levels in Vero cells treated with RSL3 at the adhesion and invasion stages. (**C**) Relative viral RNA levels in and out of Vero cells treated with RSL3 at the replication and release stages. The data were performed from three independent experiments, with the untreated group as the control group. The differences were evaluated using ANOVA test. The error bar represents the standard deviation. * *p* < 0.05, ns, not significant.

**Figure 5 viruses-15-02080-f005:**
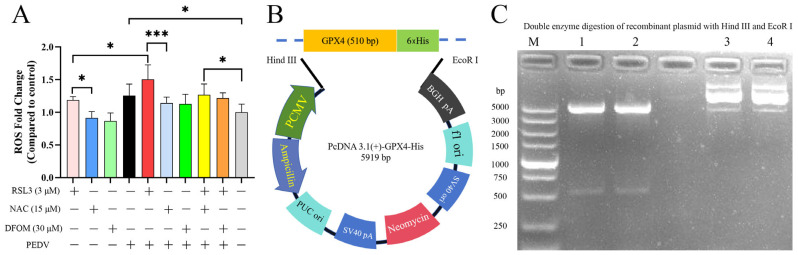
The effects of RSL3, NAC, and DFOM on ROS levels in PEDV infected Vero cells and the construction of GpX4 recombinant plasmids. (**A**) ROS levels in Vero cells treated with RSL3, NAC and DFOM for 16 h and infected with PEDV for 6 h. (**B**) pcDNA3.1-GPX4-His recombinant plasmid map. (**C**) The electrophoresis results of the recombinant plasmid pc DNA3.1-GPX4-His identified by *Hind* III and *EcoR* I double digestion showed that 1 and 2 were recombinant plasmids that were double digested, and 3 and 4 were recombinant plasmids that were not digested. The data were evaluated for differences using ANOVA test. Error bars represent standard deviation. * *p* < 0.05, *** *p* < 0.001.

**Figure 6 viruses-15-02080-f006:**
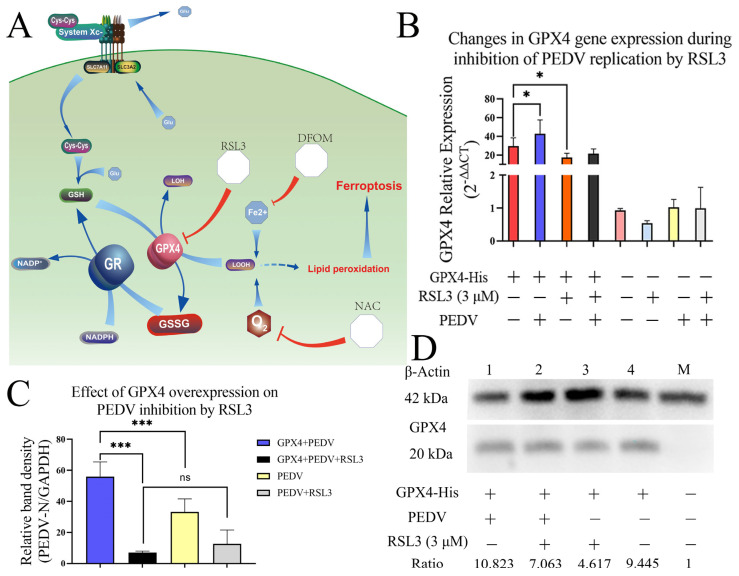
Effects of RSL3, NAC and DFOM on GPX4 expression and PEDV replication in Vero cells. (**A**) Schematic diagram of the ferroptosis pathway. (**B**) Relative GPX4 mRNA levels in Vero cells treated with RSL3, NAC and DFOM for 6 h and infected with PEDV for 6 h. (**C**) Relative PEDV RNA levels in Vero cells transfected with pcDNA3.1-GPX4-His plasmid and treated with RSL3, NAC and DFOM for 6 h and infected with PEDV for 6 h. (**D**) Expression of GPX4-His protein in Vero cells transfected with pcDNA3.1-GPX4-His plasmid and treated with RSL3, NAC, and DFOM for 6 h or infected with PEDV for 6 h, normalized to β-actin levels to obtain relative GPX4 levels. The RT-qPCR data were performed from three independent experiments, with the untreated group as the control group. The differences were evaluated using ANOVA test. The error bar represents the standard deviation. ns, not significant, * *p* < 0.05, *** *p* < 0.001, ns, not significant.

**Table 1 viruses-15-02080-t001:** Primers for real-time PCR.

Primer Name	Nucleotide Sequence (5′-3′)	Product (bp)	GenBank Accession Number or Citation Source: PMID
PEDV N	F:ACTAATAAAGGGAATAAGGACCAG	207	OP894120 PMID: 36936772
	R:GTTAGTGGGTTCAGTCTTTGC		
GPX4	F:CAGTGAGGCAAGACCGAAGTGAAC	125	NM_001039847.3 PMID:36936772
	R:TTACTCCCTGGCTCCTGCTTCC		
GAPDH	F:CCCACTCCTCCACCTTTGAC	113	NM_002046.7 PMID: 36936772
	R:TCCACCACCCTGTTGCTGTAGC		

## Data Availability

The original contributions presented in the study are included in the article, further inquiries can be directed to the corresponding author.

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
