# Peer review of "RSL3 Inhibits Porcine Epidemic Diarrhea Virus Replication by Activating Ferroptosis"

_viruses, 2023, doi:10.3390/v15102080_

Round 1

Reviewer 1 Report

The work written by Li et al and entitled "RSL3 Inhibits Porcine Epidemic Diarrhea Virus Replication by Activating Ferroptosis" is a work that describes more information about PEDV-induced pyroptosis in vitro and potential substances that act as viral promoters and inhibitors. The work, although it presents interesting results, is not currently publishable for various reasons. The authors have made numerous formatting and punctuation errors during submission. There are points in the article that need improvement (listed later). Furthermore, I am not convinced by the western blot images indicated as supplementary file. In fact, the membrane appears cut and the authors cannot demonstrate whether the two pieces of membrane actually belong to the same blot.

However, I am sure that the authors by refining the manuscript and submitting an additional western blot image (or eliminating any reference to that image) can improve the manuscript making it acceptable for publication. My specific comments are below.

Title, Line 2: Authors must specify what RLS3 is.

Line 11: Change “indice” to “induces”

Keywords: “Acetylcysteine” not suitable as keywords for this manuscript.

Line 27: Further information about virus is required. For example about the taxonomy, the history, the structure and the epidemiology (PEDV is nowadays spread throughout the world, including Europe and U.S).  I propose to the authors a review and two articles that talk about the spread of PEDV among pigs and wild boars in Europe. https://doi.org/10.1016/j.virusres.2020.198045; https://doi.org/10.1080/01652176.2022.2079756; DOI: 10.7589/JWD-D-21-00196; doi.org/10.3390/v15020300.

Line 31: What is the complex epidemiology of this virus? It is a virus that affects only swine and has fecal-oral transmission. Please delete this statement.

Line 35: Please, change the period.

Line 43: And in relation to other viruses?

Line 44-49: Change the formatting.

Line 50: Please delete “PEDV research” and reformulate the sentence.

Line 54: The period is missing.

Line 92: What are these compounds?

Line 143: Acronymous for PVDF, TBST and SDS-PAGE should be specified.

Line 144: “Gels were pried open, trimmed to size…”. What means?

Discussion: The discussion lacks references to PEDV and pyroptosis and, above all, to PEDV and other pathways linked to this pathway.

In general, I'm not sure if the authors respected the Viruses citation rules.

The English has some imperfections that the authors can correct themselves at this stage.

Author Response

Dear Editor and Reviewers,

Thank you for the professional review of our manuscript. We have carefully revised the manuscript according to the reviewer comments. Controversial, incorrect, and/or incomplete statements have been addressed and revised on a case-by-case basis. A summary of the changes made in response to each comment is provided below:

Reviewer 1 Comment:

  1. Revise the manuscript and resubmit new Western Blot images.

Response: Per your suggestion, we have repeated the experiments and submitted new Western Blot images.

  1. Title, line 2:The authors must specify what RSL3 is.

Response: Thank you for the feedback on the title wording. RSL3 is an abbreviation that we referenced from other RSL3-related article titles, which we believed represented a consensus acronym in the field. During revision, we also considered a more conclusive title such as “The ferroptosis inducer RSL3 targeting GPX4 inhibits PEDV replication in Vero cells.” However, our results can only confirm a close relationship with the ferroptosis pathway and cannot exclude additional RSL3 effects on PEDV through pathways other than GPX4 regulation. We feel the proposed title may be misleading in implying RSL3 acts solely through GPX4 and ferroptosis. Therefore, we have not changed the manuscript title.

  1. Line 11: Change "indice" to "induces"

Response: This has been changed in the manuscript.

  1. Keywords: "Acetylcysteine" is not suitable as a keyword for this manuscript.

Response: "Acetylcysteine" has been removed as a keyword.

  1. Line 27: Further information about the virus is required.

Response: Thank you for the literature recommendations, which allowed us to further improve the background. The revised background description is as follows:

Porcine epidemic diarrhea virus (PEDV) is a highly contagious coronavirus that causes severe diarrhea and mortality in neonatal piglets, resulting in substantial eco-nomic impacts on the swine industry worldwide [1,2]. PEDV is transmitted via the fe-cal-oral route and causes atrophy and shedding of small intestinal villi, resulting in se-vere diarrhea, vomiting and dehydration[3]. Effective prevention and control of PEDV remains challenging due to the virus' rapid transmission, genetic diversity, and com-plex epidemiology [4] . PEDV is distributed in many countries and regions including the United States, Europe and Asia, but PEDV in China exhibits high recombination and evolution rates, and the risk of cross-species transmission has increased, which poses challenges for our research on how to control this infectious diseas[5,6]. To reveal its pathogenesis, many researchers have made efforts to explore the interactions be-tween PEDV and many important host signaling pathways. Among them, the interac-tion between PEDV and reactive oxygen species (ROS) has also been elucidated [7-9].

  1. Line 31: What is the complex epidemiology of this virus? It is a virus that affects only swine and has fecal-oral transmission. Please delete this statement.

Response: The unreasonable phrasing has been deleted and the background paragraph has been rewritten.

  1. Line 35: Please, change the period.

Response: The punctuation has been corrected throughout the manuscript.

  1. Line 43: And in relation to other viruses?

Response: In this background section, we wanted to express that GPX4 has been reported in studies of other viruses, to support the significance of studying the GPX4 gene. We have added the transition sentence "GPX4 is one of the key influencing factors in the replication process of some viruses."

  1. Line 43: And in relation to other viruses?

Response: Apologies for this oversight, the manuscript has been modified accordingly.

  1. Line 50: Please delete “PEDV research” and reformulate the sentence.

Response: The unreasonable phrasing has been modified in the manuscript: "Recent studies have shown that lauric acid (LA) and salinomycin stimulate ROS production and inhibit PEDV replication, and both compounds are known to stimulate ROS production in cells, which is one of main characteristics of ferroptosis.".

  1. Lcine 54: The period is missing.

Response:  This has been corrected and the entire manuscript proofread.

  1. Line 92: What are these compounds?

Response: "RSL3, DFOM and NAC" has replaced "compounds".

  1. Line 143: Acronymous for PVDF, TBST and SDS-PAGE should be specified.

Response: The full names have been added at the first mention in the manuscript.

  1. Line 144: “Gels were pried open, trimmed to size…”. What means?

Response: It is our habit to remove the edges of the gel that do not contain the protein of interest before transferring. I believe this sentence can remain as is.

  1. Discussion: The discussion lacks references to PEDV and pyroptosis and, above all, to PEDV and other pathways linked to this pathway.

Response: This study mainly investigates the ferroptosis pathway, so pathways that can cause ER stress and ROS regulation were the main focus. We also examined research on PEDV and pyroptosis, particularly NLRP3 studies, but felt expanded discussion did not fit within the scope of this paper. However, we have improved the other discussion content in the manuscript.

We have carefully proofread the entire manuscript and figures. The revised manuscript has been submitted. Please review for the next steps.

Thank you for your time handling our manuscript. I look forward to hearing from you.

Sincerely,

Yingguang Li 

October 3rd, 2023

Reviewer 2 Report

The manuscript titled “RSL3 Inhibits Porcine Epidemic Diarrhea Virus Replication by Activating Ferroptosis”. The manuscript investigates the antiviral mechanism of RSL3 in Vero cells infected with Porcine Epidemic Diarrhea Virus (PEDV), the effect of RSL3 on inhibiting PEDV, and the regulation of the GPX4 gene on PEDV. Here are a few questions that need to be addressed before the manuscript can be considered for publication:

There are some unclear expressions in the manuscript. In the discussion section, it is stated: “PEDV research, lauric acid (LA) and salinomycin have been found to stimulate ROS and inhibit PEDV replication.” This needs to be elaborated further.

In the methods section, different parts of the article use different methods, such as ANOVA, t-tests, etc. These methods have different applicable conditions and need to meet certain requirements. Please clarify the analysis methods used for the experiments.

When discussing the mechanism of action of RSL3, only the perspective of GPX4 is analyzed, and other possible pathways are not considered. Whether RSL3 inhibits PEDV solely by down-regulating GPX4 or there are other pathways is not clear in the current study. You should further review related literature and discuss various possible mechanisms, which will also guide subsequent research.

The article describes the research results in detail, but it would be beneficial to provide some insights or expectations for future research.

Please address these issues in your revised manuscript.

Author Response

Thanks for the professional review of our manuscript. We have carefully revised the manuscript in accordance with the comments of Reviewers. Controversial, incorrect, and/or incomplete statements in the manuscript have been revised on a case-by-case basis in accordance with the comments. Each change made to the comment is summarized below:

Reviewer Comment:

  1. There are some unclear expressions in the manuscript. In the discussion section, it is stated: “PEDV research, lauric acid (LA) and salinomycin have been found to stimulate ROS and inhibit PEDV replication.” This needs to be elaborated further..

Response: This statement has been modified in the manuscript: "Recent studies have shown that lauric acid (LA) and salinomycin stimulate ROS production and inhibit PEDV replication, and both compounds are known to stimulate ROS production in cells, which is one of the main characteristics of ferroptosis.".

  1. In the methods section, different parts of the article use different methods, such as ANOVA, t-tests, etc. These methods have different applicable conditions and need to meet certain requirements. Please clarify the analysis methods used for the experiments.

Response: Regarding the description of statistical methods, we have confirmed the methods used. One-way ANOVA analysis was performed for comparisons between groups. "Differences among multiple groups were analyzed by ordinary one-way ANOVA followed by comparisons between each two groups."

  1. When discussing the mechanism of action of RSL3, only the perspective of GPX4 is analyzed, and other possible pathways are not considered. Whether RSL3 inhibits PEDV solely by down-regulating GPX4 or there are other pathways is not clear in the current study. You should further review related literature and discuss various possible mechanisms, which will also guide subsequent research.

Response: Thank you for the suggestion. The discussion of the RSL3 mechanism in this manuscript was indeed insufficient. We have expanded the discussion in the text, but were limited in proposing innovative perspectives beyond existing RSL3 research.

  1. The article describes the research results in detail, but it would be beneficial to provide some insights or expectations for future research.

Response: Thank you for the literature recommendations. In the conclusion, we have added possible future research directions based on this study.

We have carefully checked the writing of the entire manuscript, and We have submitted the revised manuscript and figures. Please review our manuscript for the next step.

Thank you for your time handling our manuscript. I am looking forward to hearing from you.

Sincerely,

Yingguang Li

October 3rd, 2023

Reviewer 3 Report

In this study, the authors used the ferroptosis pathway downstream target activator (1S,3R)-RSL3 compound as a starting point, combined with the interactions of N-acetylcysteine and deferoxamine, to elucidate the effects of a series of compounds on PEDV proliferation. This study belongs to the study of the molecular mechanism of viral replication in cells, which has some basic theoretical value. However, the quality of the article is not high and needs further processing and improvement. In particular, some figure notes are poorly labelled, making it impossible to read or understand the authors' results.

There are many issues with punctuation and font in this text. For example:

Line 35: the period

Line 40: also period

Line 44-50: the font is too large

Abbreviations in the Figures lack full name notes.

Note that on the right side of subfigure B in Figure 2, the grouping labels are not visible and therefore cannot be assessed.

Is there randomisation or equality between groups in the selection of fields of view in the microscope of Figure 3?

In Figure 5C, what does each of lanes 1, 2, 3, and 4 represent?

The labelling of significance between groups in all figures is a bit messy. It is not possible to read the graphs because there are more subgroups in the graphs. It is recommended that letters be used to label significance.

Author Response

Thank you for the professional review of our manuscript. We have carefully revised the manuscript according to the reviewer comments. Controversial, incorrect, and/or incomplete statements have been addressed and revised on a case-by-case basis. A summary of the changes made in response to each comment is provided below:

Reviewer Comment:

  1. There are many issues with punctuation and font in this text.

Response: There were indeed many writing errors in the text, which have been corrected in the revised manuscript.

  1. Abbreviations in the Figures lack full name notes.

Response: The full names have been added at the first mention of abbreviations in the text, and the full names of abbreviations in the figures have been completed in the figure legends.

  1. Note that on the right side of subfigure B in Figure 2, the grouping labels are not visible and therefore cannot be assessed.

Response: We have resubmitted Figure 2 to address any potential issues with image quality and unclear labels, so please let us know if they remain a problem.

  1. Is there randomisation or equality between groups in the selection of fields of view in the microscope of Figure 3?

Response: We believe the selection of fields of view in Figure 3 followed principles of authenticity and equalization. As you mentioned, there may be subjective differences in fluorescence PCR field selection. We aimed to present our observed results through image statistics and field selection as best as possible. This was straightforward for groups with significant differences, but more difficult for groups with similar lesion levels, which is why we provide two fields per group for reader assessment.

  1. In Figure 5C, what does each of lanes 1, 2, 3, and 4 represent?

Response: In Figure 5C, lanes 1 and 2 are bands after double digestion of the recombinant plasmid, and lanes 3 and 4 are bands without plasmid digestion. The purpose of this figure was to match the plasmid diagram, vividly demonstrating our experimental logic for GPX4 validation. The missing lane labels have been completed in the figure annotation.

  1. The labelling of significance between groups in all figures is a bit messy. It is not possible to read the graphs because there are more subgroups in the graphs. It is recommended that letters be used to label significance.

Response: The presentation of results in the figures was compact and not reader-friendly. We have made adjustments in the newly submitted manuscript. Each result provided guidance for subsequent experiments, so showing the complete experimental logic through the results was important. To better present significance analysis, we still used symbols but optimized the figures and legends.

We have carefully checked the writing of the entire manuscript, and We have submitted the revised manuscript and figures. Please review our manuscript for the next step.

Thank you for your time handling our manuscript. I am looking forward to hearing from you.

Sincerely,

Yingguang Li

October 3rd , 2023

Round 2

Reviewer 1 Report

The authors addressed a good portion of my comments. The manuscript has improved significantly and is almost ready for acceptance. However, the authors should submit the entire image of the new blot as a supplementary image (the old one is still present), demonstrating that this time the blot has not been cut and it is possible to observe both the normalizer and the protein under examination on the same membrane (it would also be advisable to indicate the molecular weight of the ladder). Thi is a major revision in my opinion. Below, I recommend a couple of works that can also help explain the role of wild boar in the transmission of this infection. This information could help improve the introduction. DOI: 10.7589/JWD-D-21-00196; doi.org/10.1007/s10393-022-01591-x; doi.org/10.3390/v15020300. 

English is understandable, and technical terms are used appropriately.

Author Response

Thank you for the professional review of our manuscript. Controversial, incorrect, and/or incomplete statements have been revised or explained on a case-by-case basis in accordance with the comments. A summary of the changes made in response to each comment is provided below:

Reviewer Comment:

  1. The authors should submit the entire image of the new blot as a supplementary image (the old one is still present), demonstrating that this time the blot has not been cut and it is possible to observe both the normalizer and the protein under examination on the same membrane (it would also be advisable to indicate the molecular weight of the ladder).

Response: Thank you for the suggestion raised in the first round of review regarding the issues in the manuscript. Therefore, we performed new immunoblotting on the grouping samples to obtain new images. However, during the operation, the two proteins were incubated based on different antibodies and exposed for different durations due to limitations (e.g. overexposure of β-Actin when exposing GPX4-His), as well as our operation process. Therefore, we cannot provide pictures showing the two proteins displayed on the same complete membrane. We can only provide the captured images in the supplementary material to demonstrate that the results obtained from one experiment came from one electrophoresis.

  1. Modify the introduction of the article according to DOI:10.7589/JWD-D-21-00196; 10.1007/s10393-022-01591-x; 10.3390/v15020300.

Response: We have consulted the articles recommended by you, and understand that you want us to add an introduction to the epidemiology of PEDV. In the text, we cited the key points of the articles and made a brief improvement. “PEDV is distributed in many countries and regions including the United States, Europe and Asia, but PEDV in China exhibits higher recombination and evolution rates with increased risks of cross-species transmission [5,6]. Recent epidemiological studies in southern Italy reveal variability in PEDV epidemiology and transmission risks between different animal populations[7,8]. In domestically raised pigs in intensive farming sys-tems, PEDV seroprevalence was much higher than wild boars in the same region, reaching 14.8%, and showed an increasing trend in adult pigs[8,9].” 

Thank you for your time handling our manuscript. I am looking forward to hearing from you.

Sincerely,

Yingguang Li

October 8rd, 2023

Reviewer 3 Report

The authors have basically addressed my concerns.  However, most of the experimental results in this paper are bar graphs based on assay data, and it would be better to increase the number of sample replicates and quality of immunofluorescence and Western blot, as well as in vivo validation experiments.

The quality of the article has improved considerably after the author's revisions, although there are still some writing issues that need to be refined.

Prior to publication, authors need to carefully check and reduce obvious writing problems to increase readability.

For example:

Line 226: "CFor the"

Line 291: "WESTERN Blot"

Line 358-373: Does this paragraph as a whole need to be in bold font?

Author Response

Thank you for the professional review of our manuscript. Controversial, incorrect, and/or incomplete statements have been revised or explained on a case-by-case basis in accordance with the comments. A summary of the changes made in response to each comment is provided below:

Reviewer Comment:

  1. Most of the experimental results in this paper are bar charts based on measured data. It would be better to increase sample repeats and validate the quality of immunofluorescence and western blot, as well as in vivo verification experiments.

Response: Sufficient preliminary experiments were conducted prior to obtaining the formal results presented here, which meet statistical verification. The experiments are repeatable and the results are reliable. In vivo modeling would certainly be the most effective way to validate the findings, but based on current conditions, we still need to improve certain theories and screen for drugs with greater clinical significance under this theory before conducting in vivo tests, which is work to be done going forward.

  1. Writing issues:

Line 226: "CFor the"

Line 291: "WESTERN Blot"

Lines 358-373: Does this paragraph as a whole need to be in bold font?

Response: The writing issues have been corrected again.

Thank you for your time handling our manuscript. I am looking forward to hearing from you.

真诚地

应光 Li

十月8rd,2023